# Identification of highly-protective combinations of *Plasmodium vivax* recombinant proteins for vaccine development

Camila Tenorio França[1,2], Michael T White[1,3], Wen-Qiang He[2,4],
Jessica B Hostetler[5,6], Jessica Brewster[4], Gabriel Frato[7], Indu Malhotra[7],
Jakub Gruszczyk[4], Christele Huon[8], Enmoore Lin[9], Benson Kiniboro[9],
Anjali Yadava[10], Peter Siba[9], Mary R Galinski[11,12], Julie Healer[2,4],
Chetan Chitnis[8,13], Alan F Cowman[2,4], Eizo Takashima[10], Takafumi Tsuboi[14],
Wai-Hong Tham[2,4], Rick M Fairhurst[6], Julian C Rayner[5], Christopher L King[7],
Ivo Mueller[1,2,15,16]*

[1]Division of Population Health and Immunity, Walter and Eliza Hall Institute, Parkville, Australia; [2]Department of Medical Biology, University of Melbourne, Parkville, Australia; [3]MRC Center for Outbreak Analysis and Modelling, Department of Infectious Disease Epidemiology, Imperial College London, London, United Kingdom; [4]Division of Infection and Immunity, Walter and Eliza Hall Institute, Parkville, Australia; [5]Malaria Programme, Wellcome Trust Sanger Institute, Hinxton, United Kingdom; [6]Laboratory of Malaria and Vector Research, National Institute of Allergy and Infectious Diseases, National Institutes of Health, Rockville, United States; [7]Center for Global Health and Diseases, Case Western Reserve University, Cleveland, United States; [8]Malaria Parasite Biology and Vaccines Unit, Institut Pasteur, Paris, France; [9]Malaria Immuno-Epidemiology Unit, PNG Institute of Medical Research, Yagaum, Papua New Guinea; [10]Malaria Vaccine Branch, Walter Reed Army Institute of Research, Silver Spring, United States; [11]International Center for Malaria Research, Education, and Development, Emory Vaccine Center, Yerkes National Primate Research Center, Emory University, Atlanta, United States; [12]Infectious Diseases Division, Department of Medicine, Emory University, Atlanta, United States; [13]International Centre for Genetic Engineering and Biotechnology, New Delhi, India; [14]Division of Malaria Research, Proteo-Science Center, Ehime University, Matsuyama, Japan; [15]Malaria Parasites and Hosts Unit, Department of Parasites and Insect Vectors, Institut Pasteur, Paris, France; [16]Barcelona Institute of Global Health, Barcelona, Spain

*For correspondence:
mueller@wehi.edu.au

Competing interests: The authors declare that no competing interests exist.

**Abstract** The study of antigenic targets of naturally-acquired immunity is essential to identify and prioritize antigens for further functional characterization. We measured total IgG antibodies to 38 *P. vivax* antigens, investigating their relationship with prospective risk of malaria in a cohort of 1–3 years old Papua New Guinean children. Using simulated annealing algorithms, the potential protective efficacy of antibodies to multiple antigen-combinations, and the antibody thresholds associated with protection were investigated for the first time. High antibody levels to multiple known and newly identified proteins were strongly associated with protection (IRR 0.44–0.74, p<0.001–0.041). Among five-antigen combinations with the strongest protective effect (>90%), EBP, DBPII, RBP1a, CyRPA, and PVX_081550 were most frequently identified; several of them

requiring very low antibody levels to show a protective association. These data identify individual antigens that should be prioritized for further functional testing and establish a clear path to testing a multicomponent *P. vivax* vaccine.

DOI: https://doi.org/10.7554/eLife.28673.001

## Introduction

*Plasmodium vivax* is the predominant species causing malaria in the Americas and Asia-Pacific regions (*WHO, 2014*). Due to its unique biology, *P. vivax* is less susceptible to commonly-used vector control measures (*Mueller et al., 2009*). With renewed funding and commitment towards the goal of eliminating malaria, the development and deployment of an effective vaccine against *P. vivax* has become a high priority (*malERA Consultative Group on Vaccines, 2011*).

The epidemiology of *P. vivax* provides strong indication that the development of an effective vaccine is feasible. Clinical immunity to *P. vivax* is acquired more rapidly than to *P. falciparum* and, in moderate and highly endemic areas, clinical disease is virtually absent in children older than five years (*Phimpraphi et al., 2008*; *Lin et al., 2010*; *Robinson et al., 2015*). Sub-microscopic infections are common even in areas where malaria transmission has been substantially reduced (*Cheng et al., 2015*; *Bousema et al., 2014*), suggesting the development of effective immunity that provides good control of parasitemia. Whilst our understanding of the protective immune effector mechanisms is incomplete, the humoral component is thought to be essential for the development of anti-*P. vivax* immunity, and antibodies confer protection by both preventing high-density parasitemia and inhibiting blood-stage replication (*Longley et al., 2016*). Evidence of relatively long-lived IgG levels and/or seropositivity to *P. vivax* proteins such as circumsporozoite protein (CSP), merozoite surface protein 1 (MSP1), apical membrane antigen 1 (AMA1) and Duffy binding protein (DBP) have been reported even in the absence of detectable *P. vivax* infections (*Longley et al., 2015*; *Wipasa et al., 2010*; *Achtman et al., 2005*).

A highly-efficacious malaria vaccine has unfortunately not yet been achieved for any of the *Plasmodium spp.* parasites, and the vast majority of vaccines in development are based on single recombinant protein antigens (*Halbroth and Draper, 2015*). For *P. falciparum,* more than 15 antigens are under pre-clinical or clinical testing, including the pre-erythrocytic, CSP-based RTS,S/AS01 vaccine, which has recently completed Phase 3 trials (*Agnandji et al., 2012*; *RTS,S Clinical Trials Partnership, 2014*). In contrast, for *P. vivax* only a handful of proteins have been studied, with the blood-stage DBP, MSP1, and transmission-blocking protein Pvs25 as the main vaccine candidates (*Halbroth and Draper, 2015*; *Mueller et al., 2015*). The moderate success of the *P. falciparum* RTS,S vaccine has also made *P. vivax* CSP a main vaccine candidate (*Mueller et al., 2015*; *Salman et al., 2017*; *Bennett et al., 2016*). The potential of novel, pre-erythrocytic candidate antigens, including *P. vivax* cell-traversal protein for ookinetes (CelTOS) and thrombospondin-related anonymous protein (TRAP) is being investigated in animal models (*Mueller et al., 2015*; *Alves et al., 2017*; *Bauza et al., 2014*).

Recent serological screens of large panels of *P. falciparum* proteins in naturally-exposed populations have identified several antigenic targets associated with protective immunity, and strongly support the hypothesis that a multicomponent vaccine possibly including antigens from different stages of the parasites life cycle would offer a higher degree of protection than a single component vaccine (*Halbroth and Draper, 2015*; *Richards et al., 2013*; *Osier et al., 2014a*). For *P. vivax*, the availability of the genome sequence (*Carlton et al., 2008*) and transcriptome (*Bozdech et al., 2008*) have enhanced our understanding of the parasite's biology, facilitating the identification of many proteins that are homologous to *P. falciparum* antigens (*Chen et al., 2010*; *Lu et al., 2014*; *Finney et al., 2014*). Given the lack of methods to continuously culture *P. vivax* to study specific proteins in vitro, or animal models for extensive in vivo assays, the study of targets of human natural immunity in exposed populations must play an essential role in identifying and prioritizing *P. vivax* antigens for further functional characterization for vaccine or biomarker development (*Chia et al., 2014*). To date, however, the number, identity, and relative importance of *P. vivax* antigenic targets remains poorly explored (*Cutts et al., 2014*). Antibodies to proteins such as MSP3α, MSP9 (30) and DBP (*King et al., 2008*; *Cole-Tobian et al., 2009*) have shown strong associations with reduced risk of

clinical disease or blood-stage infection. Screening studies to extend the pool of *P. vivax* vaccine candidates remain scarce (*Cutts et al., 2014*).

To explore the reservoir of antigenic targets and prioritize candidates for functional characterization, we investigated the association between IgG to a comprehensive library of 38 *P. vivax* recombinant antigens and risk of vivax malaria in a cohort of young Papua New Guinean (PNG) children with well-characterized differences in exposure to *P. vivax* infections (*Lin et al., 2010*). To our knowledge, the potential existence of synergistic or additive effects of combinations of antibody responses to a large panel of *P. vivax* antigens has never been explored. We applied a novel simulated annealing algorithm to efficiently explore the vast space of antigen combinations and thus identify combinations with optimal potential protective efficacy (PPE). High antibody levels to multiple antigens, including several novel proteins were strongly associated with reduced risk of vivax malaria, independently of individual differences in exposure, age, and transmission season. Combinations of antigens were identified with a PPE of >90%. These data identify several antigens that should be prioritized for further functional characterization. The naturally-acquired anti-malarial immunity explored in this study supports the development of a highly-efficacious, multicomponent *P. vivax* vaccine.

## Results

### Breadth of anti-*P. vivax* IgG antibodies in young PNG children

We selected 38 *P. vivax* antigens that are predominantly expressed at the late-schizont stage and have a potential role in erythrocyte binding or invasion. A complete description of antigens and their accession numbers can be found in *Supplementary file 1*. Antigens were expressed using mostly the wheat germ cell free (WGCF) system, human embryonic kidney (HEK) 293E cells, and *Escherichia coli* (*Tsuboi et al., 2010*; *Hostetler et al., 2015*; *Gruszczyk et al., 2016*). Most were individual, little-studied antigens, but in the case of the major vaccine candidate DBPII, several alleles were included as antibody responses have been shown to be strain specific (*Cole-Tobian et al., 2009*). In addition, for one antigen (MSP3α), we included both conserved (N and C terminals) and polymorphic (block 1 and block 2) sub-domains (*Rayner et al., 2002*), as the different regions have been shown to differ in their immunogenicity and association with protective immunity (*Stanisic et al., 2013*).

Although children were reactive to all 38 proteins tested, antibody seroprevalence varied largely (*Table 1*). Whereas more than 56% of children had already acquired antibody levels to Pv-fam-a/PVX_088820 that were above 10% of the levels observed in pooled serum from PNG adults (considered to be the 'steady-state equilibrium' levels achievable under natural exposure [*França et al., 2016a*]), none of the children had achieved antibody levels above this threshold to antigens such as GAMA, PVX_081550, or P12 (median = 12.7%, interquartile range [IQR] 3.5–31.7; *Table 1*).

To study the breadth of anti-*P. vivax* antibodies, for each antigen, antibody levels were stratified into tertiles and scored as 0, 1 and 2 for the low, medium and high tertiles, respectively. Scores were then summed to yield a median score of 37 (IQR 20–55), and fitted as a continuous variable in a negative binomial generalized estimating equation (GEE) model. Antibody repertoire increased only moderately with increasing age, with the most marked increases occurring during the first two years of life (p=0.049) (*Figure 1a*).

Given the young age and large heterogeneity in exposure among children, however, age alone is not the best proxy for lifetime exposure to malaria (*Lin et al., 2010*; *Koepfli et al., 2013*). All *P. vivax* infections occurred during the follow-up of this cohort have been genotyped, and the molecular force of blood-stage infections (molFOB, i.e. the number of genetically distinct blood-stage infections acquired over time) calculated. The molFOB is therefore a direct measure of individual differences in exposure to *P. vivax* infections (*Koepfli et al., 2013*), and has demonstrated strong correlation with factors associated with increased exposure to malaria parasites such as seasonality, geographical location, and use of bed nets; and has been shown to be the major predictor of clinical disease in this cohort of children (*Koepfli et al., 2013*; *Mueller et al., 2012*). Hence, the product of molFOB and age represents a more precise estimation of lifetime exposure to *P. vivax*. As expected, children with the highest lifetime exposure to *P. vivax* infections were able to recognize a higher number of proteins (p=0.030–0.001) (*Figure 1b*). Children currently infected (detected by PCR) were

**Table 1.** Seroprevalence of antibodies to 38 *P. vivax* proteins in Papua New Guinean children aged 1–3 years.

| Location | Protein | Geom mean* | 95% CI* | | No. of children (%) | | | | |
|---|---|---|---|---|---|---|---|---|---|
| | | | | | 1% of adult levels | 5% of adult levels | 10% of adult levels | 25% of adult levels | 50% of adult levels |
| GPI-anchored merozoite surface | MSP1 19 | 0.47 | 0.38 | 0.57 | 152 (67.9) | 67 (29.9) | 38 (17.0) | 21 (9.4) | 13 (5.8) |
| | P12 | 0.02 | 0.02 | 0.03 | 33 (14.7) | 2 (0.9) | 0 | 0 | 0 |
| Peripheral surface | MSP3a full | 1.17 | 1.05 | 1.31 | 222 (99.1) | 121 (54.0) | 51 (22.8) | 12 (5.4) | 3 (1.3) |
| | MSP3a block 1 | 0.79 | 0.72 | 0.86 | 222 (99.1) | 82 (36.6) | 18 (8.0) | 3 (1.3) | 0 |
| | MSP3a block 2 | 0.54 | 0.48 | 0.60 | 202 (90.2) | 48 (21.4) | 13 (5.8) | 4 (1.8) | 1 (0.4) |
| | MSP3a N-term | 0.11 | 0.10 | 0.13 | 222 (99.1) | 139 (62.0) | 64 (28.6) | 11 (4.9) | 4 (1.8) |
| | MSP3a C-term | 0.11 | 0.10 | 0.13 | 62 (27.7) | 4 (1.8) | 1 (0.4) | 0 | 0 |
| | MSP9 N-term | 0.09 | 0.08 | 0.11 | 62 (27.7) | 7 (3.1) | 3 (1.3) | 0 | 0 |
| | P41 | 0.02 | 0.02 | 0.02 | 20 (8.9) | 8 (3.6) | 3 (1.3) | 0 | 0 |
| | SERA | 0.96 | 0.87 | 1.07 | 224 (99.6) | 93 (41.4) | 38 (16.9) | 8 (3.6) | 3 (1.3) |
| Microneme | AMA1 | 0.41 | 0.36 | 0.47 | 167 (74.6) | 34 (15.2) | 19 (8.5) | 5 (2.2) | 1 (0.4) |
| | DBPII Sal1 | 0.24 | 0.21 | 0.27 | 127 (56.7) | 17 (7.6) | 9 (4.0) | 3 (1.3) | 2 (0.9) |
| | DBPII P | 0.23 | 0.19 | 0.28 | 125 (55.8) | 26 (11.6) | 12 (5.4) | 3 (1.3) | 3 (1.3) |
| | DBPII O | 0.34 | 0.28 | 0.40 | 146 (65.2) | 38 (17.0) | 13 (5.8) | 5 (2.2) | 3 (1.3) |
| | DBPII AH | 0.24 | 0.21 | 0.27 | 128 (57.1) | 15 (6.7) | 7 (3.1) | 2 (0.9) | 2 (0.9) |
| | DBPII C | 0.23 | 0.19 | 0.27 | 125 (55.8) | 26 (11.6) | 11 (4.9) | 3 (1.3) | 3 (1.3) |
| | EBP | 0.40 | 0.33 | 0.48 | 142 (63.1) | 60 (26.7) | 36 (16.0) | 14 (6.2) | 6 (2.7) |
| | GAMA | 0.01 | 0.01 | 0.01 | 5 (2.2) | 1 (0.4) | 0 | 0 | 0 |
| | CyRPA | 0.54 | 0.42 | 0.69 | 139 (61.8) | 78 (34.7) | 54 (24.0) | 40 (17.8) | 31 (13.8) |
| Rhoptry | ARP | 0.40 | 0.37 | 0.43 | 205 (91.1) | 17 (7.6) | 6 (2.7) | 0 | 0 |
| | RBP1a | 0.41 | 0.35 | 0.47 | 162 (72.3) | 39 (17.4) | 18 (8.0) | 9 (4.0) | 3 (1.3) |
| | RBP2a | 0.86 | 0.72 | 1.04 | 186 (83.0) | 102 (45.5) | 64 (28.6) | 29 (12.9) | 13 (5.8) |
| | RBP2b | 1.19 | 1.02 | 1.38 | 209 (93.3) | 130 (58.0) | 90 (40.2) | 12 (5.4) | 3 (1.3) |
| | RBP2cNB | 0.40 | 0.34 | 0.47 | 159 (71.0) | 40 (17.9) | 29 (12.9) | 13 (5.8) | 8 (3.6) |
| | RBP2-P2 | 1.68 | 1.49 | 1.91 | 224 (100.0) | 156 (69.6) | 89 (39.7) | 24 (10.7) | 13 (5.8) |
| | RhopH2 | 1.40 | 1.26 | 1.57 | 224 (99.6) | 144 (64.0) | 72 (32.0) | 18 (8.0) | 3 (1.3) |
| | RAMA | 1.44 | 1.30 | 1.61 | 225 (100.0) | 146 (64.9) | 61 (27.1) | 20 (8.9) | 7 (3.1) |
| Pre-erythrocytic | CSP | 0.15 | 0.12 | 0.18 | 95 (42.4) | 21 (9.4) | 8 (3.6) | 2 (0.9) | 1 (0.4) |
| | PVX_080665 | 0.68 | 0.61 | 0.76 | 214 (95.5) | 59 (26.3) | 28 (12.5) | 4 (1.8) | 1 (0.4) |
| Other | PVX_081550 | 0.03 | 0.03 | 0.04 | 6 (2.7) | 0 | 0 | 0 | 0 |
| | PVX_094350 | 1.44 | 1.30 | 1.59 | 225 (100.0) | 148 (65.8) | 65 (28.9) | 15 (6.7) | 6 (2.7) |
| | AKLP2 | 1.35 | 1.20 | 1.52 | 225 (100.0) | 134 (59.6) | 71 (31.6) | 17 (7.6) | 7 (3.1) |
| | PVX_087670 | 1.71 | 1.54 | 1.89 | 225 (100.0) | 160 (71.1) | 80 (35.6) | 22 (9.8) | 7 (3.1) |
| | PVX_122805 | 2.04 | 1.85 | 2.24 | 225 (100.0) | 189 (84.0) | 108 (48.0) | 24 (10.7) | 6 (2.7) |
| | CCp5 | 1.69 | 1.52 | 1.88 | 225 (100.0) | 162 (72.0) | 79 (35.1) | 20 (8.9) | 9 (4.0) |
| | PVX_114330 | 2.16 | 1.98 | 2.37 | 225 (100.0) | 198 (88.0) | 120 (53.3) | 28 (12.4) | 4 (1.8) |
| | Pv-fam-a/ PVX_088820 | 2.38 | 2.17 | 2.60 | 225 (100.0) | 209 (92.9) | 127 (56.4) | 32 (14.2) | 6 (2.7) |
| | Pv-fam-a/ PVX_092995 | 1.85 | 1.70 | 2.02 | 225 (100.0) | 184 (81.8) | 93 (41.3) | 18 (8.0) | 5 (2.2) |

Abbreviations: No = number; Geom mean = geometric mean; 95% CI = 95% confidence interval. *IgG levels multiplied by 1000. Values are in relative antibody units interpolated from standard curves using a 5PL logistic regression model.

DOI: https://doi.org/10.7554/eLife.28673.002

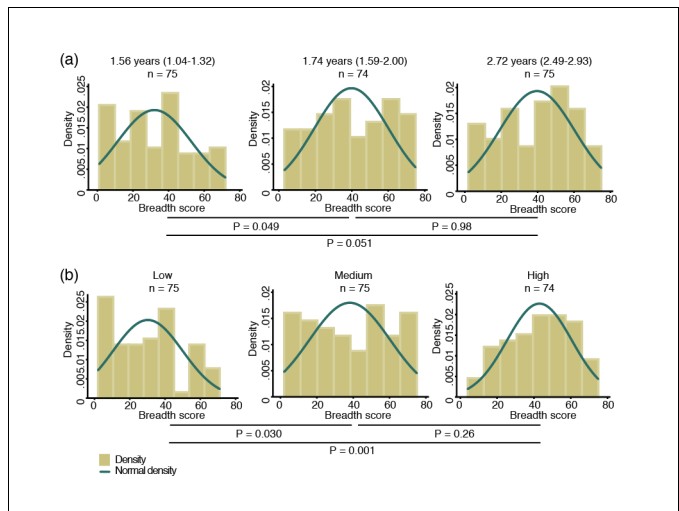

**Figure 1.** Breadth of IgG antibodies to 38 *P. vivax* proteins in Papua New Guinean children aged 1–3 years. For each protein, antibody levels were stratified into tertiles and scored as 0, 1 or 2 for the low, medium, and high tertiles, respectively. Scores were then added up to reflect the breadth of antibodies per child. (a) Antibody breadth by age group. Age is shown as median (interquartile range). (b) Antibody breadth by lifetime exposure group. For each child, exposure was defined as the total number of *P. vivax* blood-stage clones acquired per time-at-risk (molFOB), and lifetime exposure as a product of age and molFOB. P values are from negative binomial regression and were deemed significant if <0.05.

DOI: https://doi.org/10.7554/eLife.28673.003

also able to recognize more proteins (median breadth score = 43, IQR 26–60) than those that were infection-free at the time of sampling (median = 31, IQR 12–45, p<0.001).

## Exposure-related changes in magnitude of IgG levels

The overall antibody levels (i.e. the sum of antibodies to all proteins per child) also increased moderately with age (Pearson correlation coefficient = 0.15, p=0.019) (*Figure 2a*). Increases in antibody levels with age were observed for 22 individual antigens (Spearman's rho = 0.14–0.28, P<0.001–0.031) (*Figure 2a*; *Figure 2—source data 1*). Our proxy for lifetime exposure was a substantially better predictor of antibody levels than age, and overall IgG levels increased more strongly with lifetime exposure (Pearson correlation coefficient = 0.24, p<0.001) (*Figure 2b*). IgG to 29 individual proteins increased with lifetime exposure (rho = 0.14–0.38, p<0.001–0.040), the majority of them more strongly in children with a current *P. vivax* infection (rho = 0.20–0.43, P<0.001–0.048) (*Figure 2b*; *Figure 2—source data 1*).

## IgG levels and prospective risk of vivax malaria

Over 16 months of follow-up, the subset of children included in this study experienced an incidence rate of 1.25 (95%CI 1.08–1.45) clinical episodes defined as fever or history of fever in the last 48 hr with concurrent *P. vivax* parasitemia >500/μL. After adjusting our GEE model for age, season, village of residency, and individual differences in exposure as measured by molFOB (*Koepfli et al., 2013*), high antibody levels to 31 of the 38 antigens tested (81.6%) were associated with reduced risk of vivax malaria (incidence rate ratio [IRR] high versus low tertiles of responders 0.44–0.70, p<0.001–0.035) (*Figure 3*; *Figure 3—source data 1*). When antigen-specific responses were ranked according to the strength of their protective associations, except for MSP3α, all other antigens in the top-10 were either novel or understudied proteins, including EBP, PVX_081550, PVX_122805, CyRPA, RBP1a, RBP2b and P41 (*Figure 3*).

In unadjusted analysis, children with medium and/or high IgG levels to several *P. vivax* antigens showed an increased risk of falciparum malaria with density <2500 parasites/μL (IRR 1.24–1.50, p=0.001–0.045; *Figure 3—source data 2*), suggesting that in this age group antibodies to *P. vivax* are also markers of a higher co-exposure to *P. falciparum* infections, and thus increased risk of

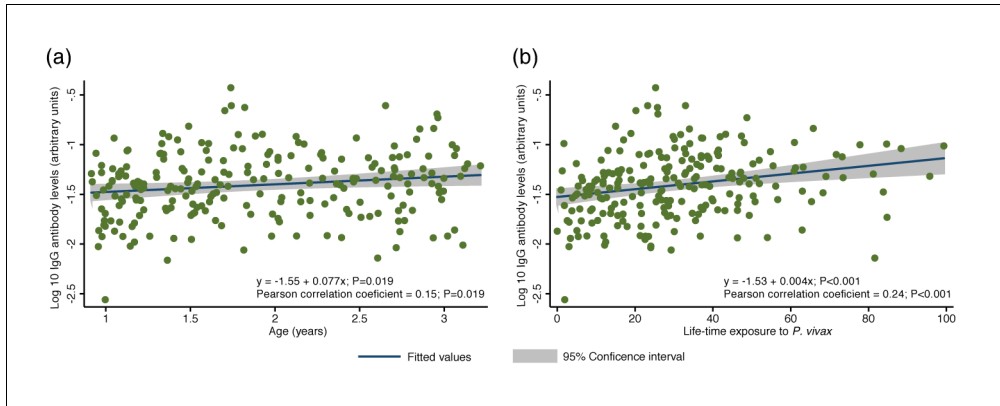

**Figure 2.** Association between cumulative IgG levels to 38 *P. vivax* proteins and exposure to *P. vivax* infections in Papua New Guinean children aged 1–3 years.  (a) Association with age. (b) Association with lifetime exposure. For each child, exposure was defined as the total number of *P. vivax* blood-stage clones acquired per time-at-risk (molFOB), and lifetime exposure as a product of age and molFOB. n = 225. P values < 0.05 were deemed significant.

DOI: https://doi.org/10.7554/eLife.28673.004

The following source data is available for figure 2:

**Source data 1.** Associations between IgG to *P. vivax* antigens with measures of concurrent and cumulative exposure in Papua New Guinean children aged 1–3 years.
DOI: https://doi.org/10.7554/eLife.28673.005

developing falciparum malaria (*Mueller et al., 2012*). After adjusting for covariates, antibodies to none of the *P. vivax* antigens were associated with reduced risk of falciparum malaria (with any parasite density), suggesting that the protection observed for *P. vivax* is species-specific (*Figure 3*; *Figure 3—source datas 1* and *2*).

## Co-acquisition and cross-reactivity of IgG levels to different proteins

Antibody responses to the 38 different antigens were, however, significantly correlated (rho = 0.14–0.995, p<0.05) (*Figure 4*). The strongest correlations were among antibodies to the different alleles of DBPII (rho = 0.76–0.995, p<0.001) (*Figure 4*). DBPII shares 50% identity with EBP, and correlations between antibodies to the different alleles of DBPII and EBP were moderate (rho = 0.36–0.48, p<0.001) (*Figure 4*). Antibody levels to the different RBPs (rho = 0.51–0.70, p<0.001) and to different regions of MSP3α were also highly correlated (rho = 0.52–0.93, p<0.001). Among the other proteins, antibodies to the WGCF-expressed, highly-immunogenic proteins generally showed very high correlations (rho = 0.61–0.95, p<0.001), while antibodies to the HEK293E-rxpressed, full-ectodomain proteins showed lower correlations (rho = 0.15–0.66, P<0.001–0.035), both among themselves and to other proteins (*Figure 4*).

## Potential protective efficacy of IgG antibodies to protein combinations

The complex and strongly-correlated nature of naturally-acquired antibodies makes prioritizing individual proteins based solely on the strength of their respective associations with malaria risk disease extremely difficult. We therefore investigated the effect of responses to combinations of proteins on the risk of vivax malaria, based on the premise that a combination of responses is likely to be more efficacious than a response to a single component, given the complexity of *P. vivax* biology and the (co-) acquisition of humoral immunity to this parasite. For this we applied a novel simulated annealing algorithm to efficiently explore the vast space of antigen combinations, examining the potential protective efficacy (PPE) of combining antibodies to 5–38 antigens on the association with risk of disease.

Association with protection from clinical malaria increased with increasing the number of antigens, but at a diminishing rate (*Figure 5a and b*). The PPE increased from <75% for individual antigens to >90% for combinations of antibodies to 5–38 proteins. Including antibody response to more than 20 proteins, however, did not lead to a significant further increase in PPE (*Figure 5a and b*).

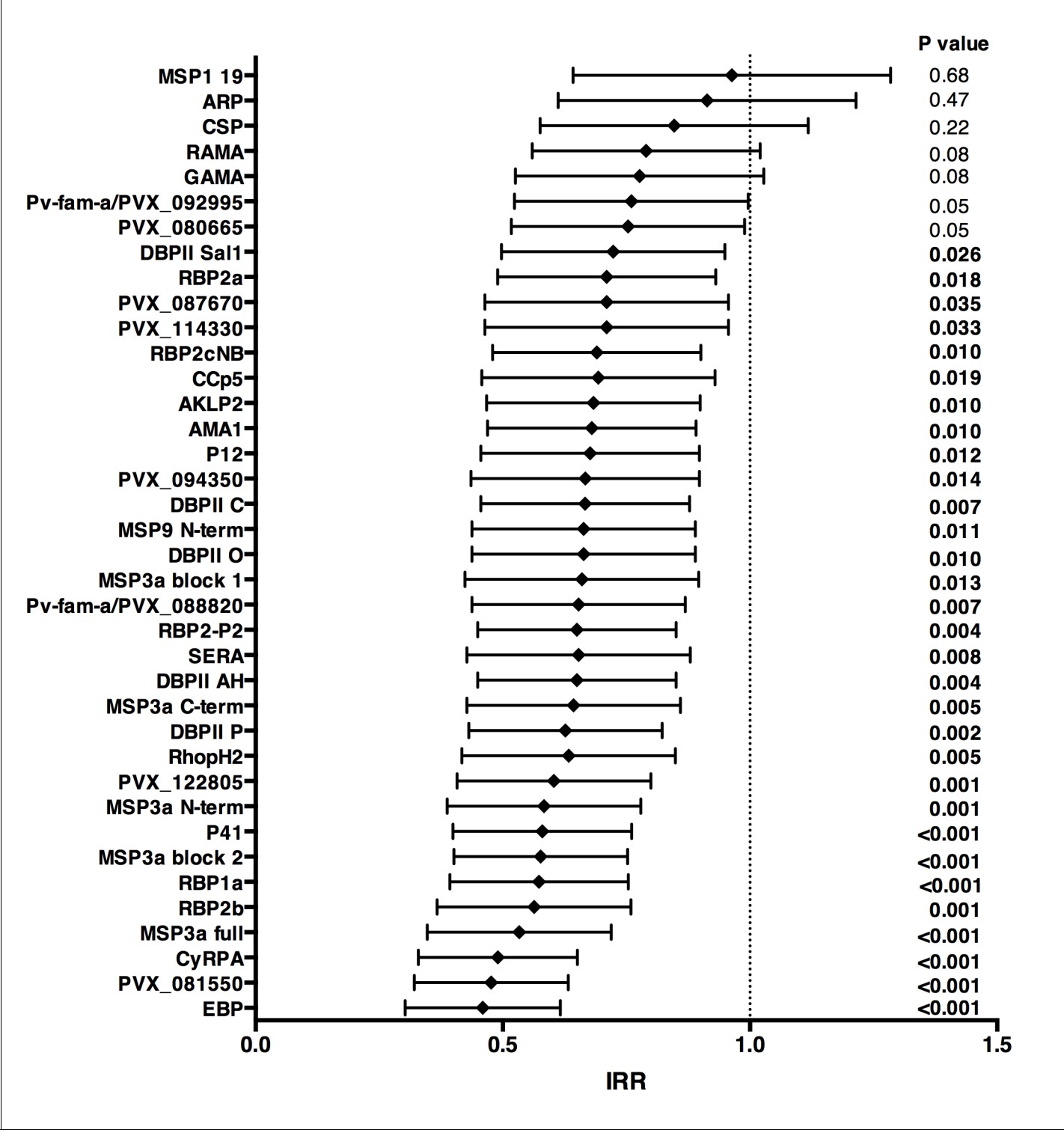

**Figure 3.** Association between high IgG levels to 38 *P. vivax* proteins and protection against clinical malaria (density >500/µL) in Papua New Guinean children aged 1–3 years old. Data are plotted as incidence rate ratios and 95% confidence intervals adjusted for exposure (molFOB), age, season, and village of residency. Incidence rate ratios are for high versus low tertiles of responders, 95% confidence intervals and P values are from GEE models. P values < 0.05 were deemed significant.

DOI: https://doi.org/10.7554/eLife.28673.006

The following source data is available for figure 3:

*Figure 3 continued on next page*

*Figure 3 continued*

**Source data 1.** Associations between antibodies to 38 *P. vivax* proteins and risk of *P. vivax* clinical episodes (>500 parasites/µL) in Papua New Guinean children aged 1–3 years.

DOI: https://doi.org/10.7554/eLife.28673.007

**Source data 2.** Associations between antibodies to 38 *P. vivax* proteins and risk of *P. falciparum* clinical episodes (>2500 parasites/µL) in Papua New Guinean children aged 1–3 years.

DOI: https://doi.org/10.7554/eLife.28673.008

Synergistic or additive effects could also be restricted to specific combinations of proteins. The PPE of every possible combination of one to five, as well as 33 to 38 proteins was investigated, and optimal five-way combinations were identified that already showed >90% protective efficacy without addition of further proteins (*Figure 5a and b*). Antigens with higher individual protective efficacies were more likely to be included in any five-component, highly-efficacious combination (*Figure 5c*

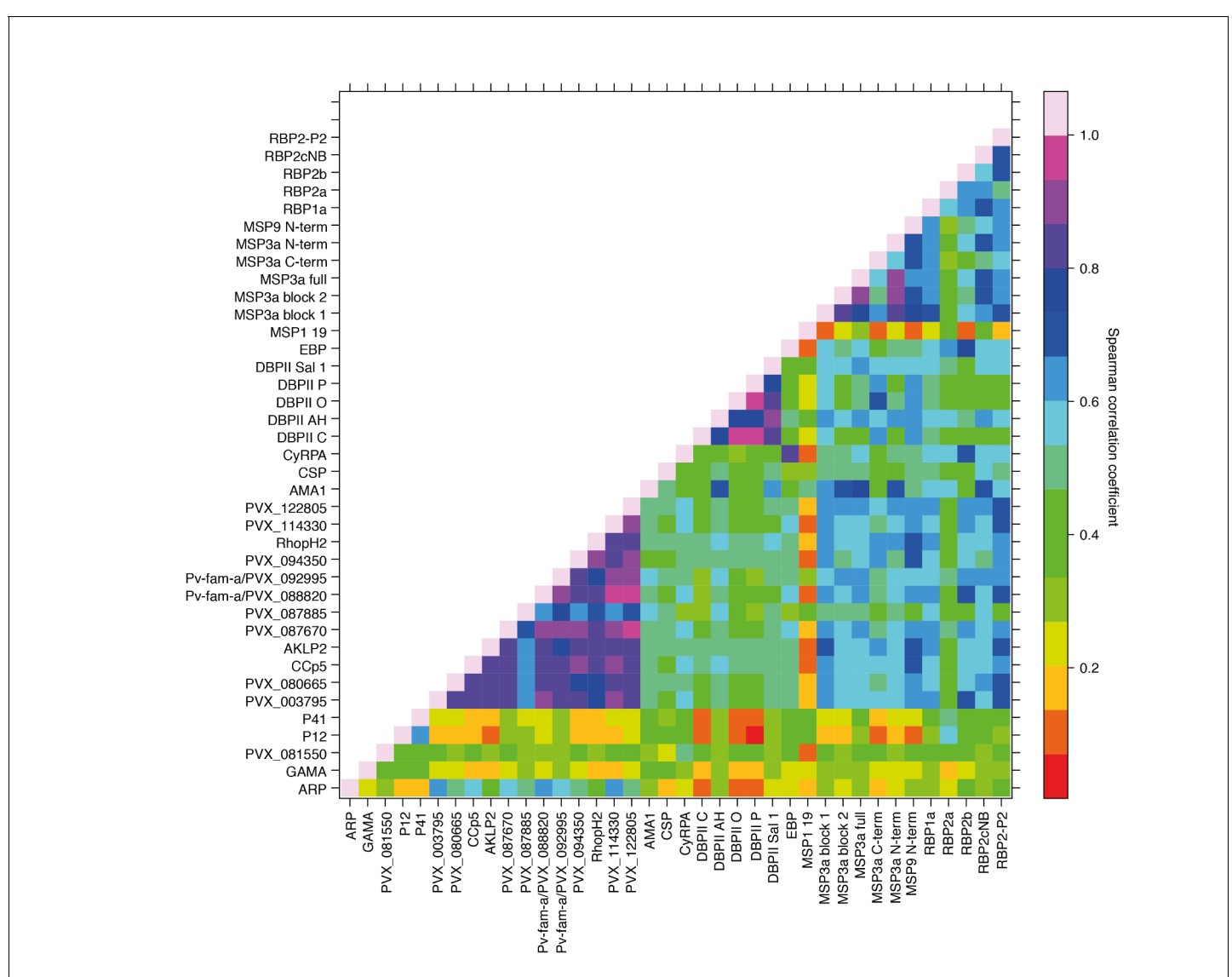

**Figure 4.** Correlations between IgG to 38 *P. vivax* proteins in Papua New Guinean children aged 1–3 years. Correlation coefficients between antibody levels to every pair of antigens were calculated using Spearman's rank correlation tests.

DOI: https://doi.org/10.7554/eLife.28673.009

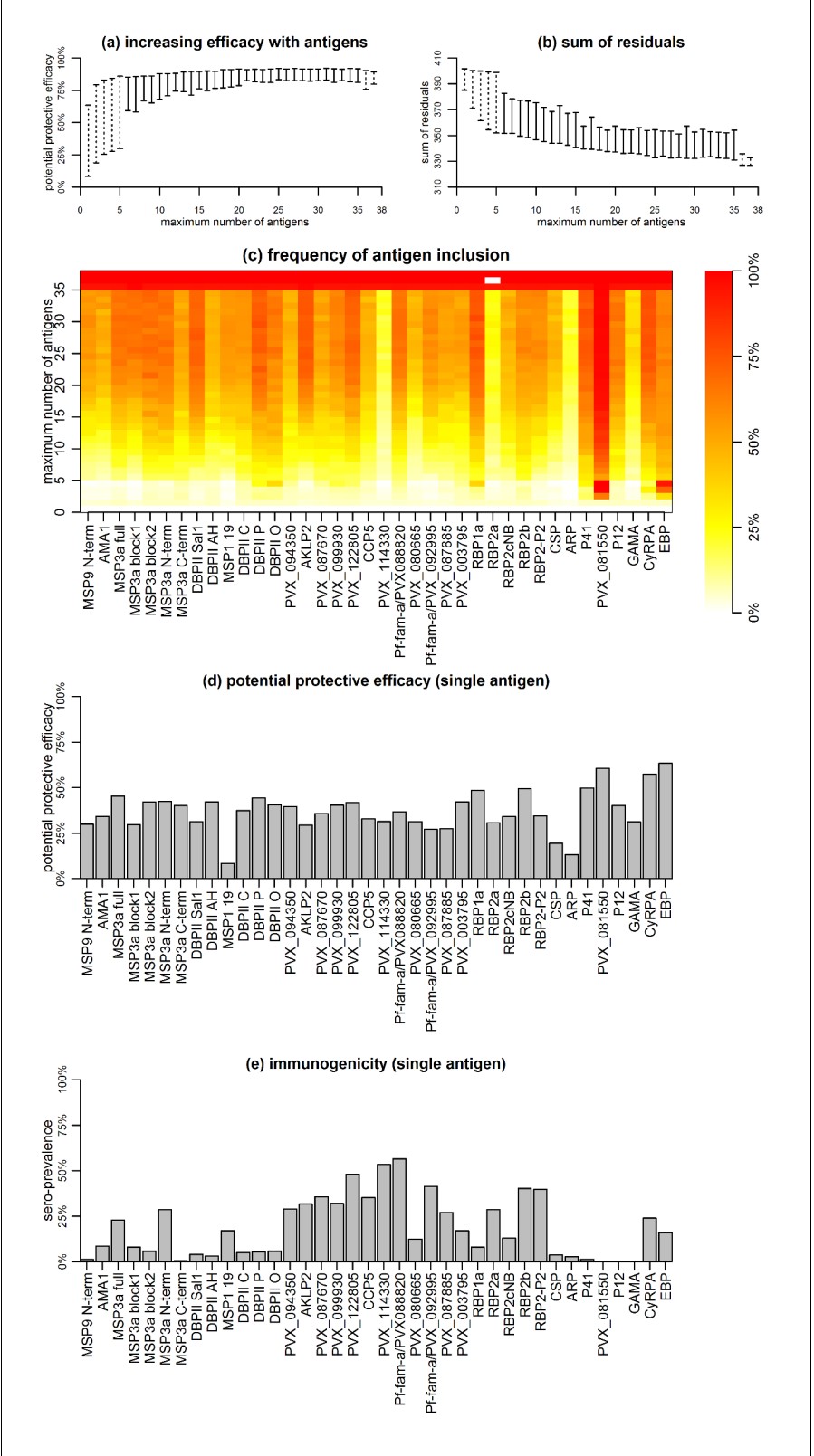

**Figure 5.** Association between antibodies to combinations of *P. vivax* proteins and malaria risk in Papua New Guinean children aged 1–3 years. (a) Potential protective efficacy (PPE) for combinations of antigens with the maximum number of antigens indicated on the x-axis. Dashed lines represent the range of PPE for all possible combinations. Solid lines represent the range of PPE from 1000 implementations of the simulated annealing algorithm. (b) Sum of residuals (as a measure of model goodness of fit). (c) The heatmap shows the frequency of including an antigen (x-axis) in a multi-component

*Figure 5 continued on next page*

*Figure 5 continued*
vaccine with a fixed number of antigens (y-axis). (**d**) Predicted PPE of a single antigen. (**e**) Immunogenicity of each antigen represented as seroprevalence with a cut-off as 10% of the antibody levels of fully-immune PNG adults.
DOI: https://doi.org/10.7554/eLife.28673.010

*and d*). These included EBP, PVX_081550, P41, DBPII (variant O), RBP1a, and CyRPA, but not MSP3α for example (*Figure 5c*). In combinations of more than five antigens, these antigens were also consistently selected (*Figure 5c*). There was no obvious evidence that the probability of inclusion was associated with higher immunogenicity of the antigen (*Figure 5e*). Among antigens with the lowest probability of being included in any combination with <35 antigens were both the highly-immunogenic PVX_114330 and the poorly-immunogenic ARP and GAMA (*Figure 5c and e*).

## Thresholds of IgG levels and protection

Although it was not possible to give an absolute quantification of antigen-specific antibody concentration in this study, there was evidence of different antigen-specific dose-response patterns and association with reduced risk of vivax malaria. Using a mathematical dose-response model adjusted for exposure (molFOB), predicted threshold ranges (relative to the PNG immune adult pool) at which antibodies become associated with protection were observed for the different antigens (*Figure 6*; *Figure 6—figure supplement 1*).

The dose-response model predicts that among all antigens, MSP3α C-term and MSP9 N-term need the lowest levels of specific antibodies to show an association with 50% reduction in symptomatic infections, followed by CyRPA, EBP, RBP1a, RBP2b, and RBP2-P2 (*Figure 6a–c*; *Figure 6—figure supplement 1*). Intermediate antibody levels were predicted for the full-ectodomain and other regions of MSP3α, as well as several of the bioinformatically-selected proteins, including CCp5, AKLP2, and PVX_087670 among others (*Figure 6d–f*; *Figure 6—figure supplement 1*). Whereas the different DBPII alleles have intermediate profiles, increasing levels of antibodies to the P, C, and O variants of DBPII were associated with a gradual reduction in the incidence of clinical disease (*Figure 6—figure supplement 1*). A 50% 'protective' effect, however, would need very high antibody levels for antigens such as Pv-fam-a/PVX_088820, SERA and PVX_114330, or may never be achieved with increasing antibody levels to RBP2a, MSP1 19, and others (*Figure 6g–l*; *Figure 6—figure supplement 1*).

In order to assure the observed results were not influenced by the concentration of the antigen conjugated onto the assay beads, we correlated the antigen concentration with the geometric mean antibody levels observed in children, the levels at which 50% protection is achieved (if it is achieved at all), and the maximum level of protection achieved. For none of these variables a significant correlation was observed (rho = −0.16 to 0.20, p>0.3), indicating that the conjugated antigen concentrations did not influence the relationship between antibody levels and the level protection achieved (*Figure 6—source data 1*).

## Discussion

The development and deployment of an effective *P. vivax* vaccine has become a priority for accelerating malaria elimination in the Asia-Pacific and the Americas (*Tanner et al., 2015*). To date, however, remarkably few *P. vivax* vaccine candidates are close to or have reached clinical trials (*Mueller et al., 2015*). To expand and prioritize the reservoir of antigenic targets for further functional characterization for use in vaccine or biomarker development, we measured total IgG antibody levels to 38 *P. vivax* recombinant proteins, investigating their relationship with prospective risk of vivax malaria in a longitudinal cohort of PNG children aged 1–3 years. For the first time, the potential protective efficacy of combination of antigens, and the thresholds necessary for an association with protection were investigated.

Although the young children included in the present study are actively acquiring clinical immunity, as indicated by a highly-significant decrease in incidence of clinical *P. vivax* episodes (*Lin et al., 2010*), most still had relatively low antibody responses, with more than 10% of children exceeding 10% of adult levels to only 22 proteins. Antibodies to almost all proteins were boosted by

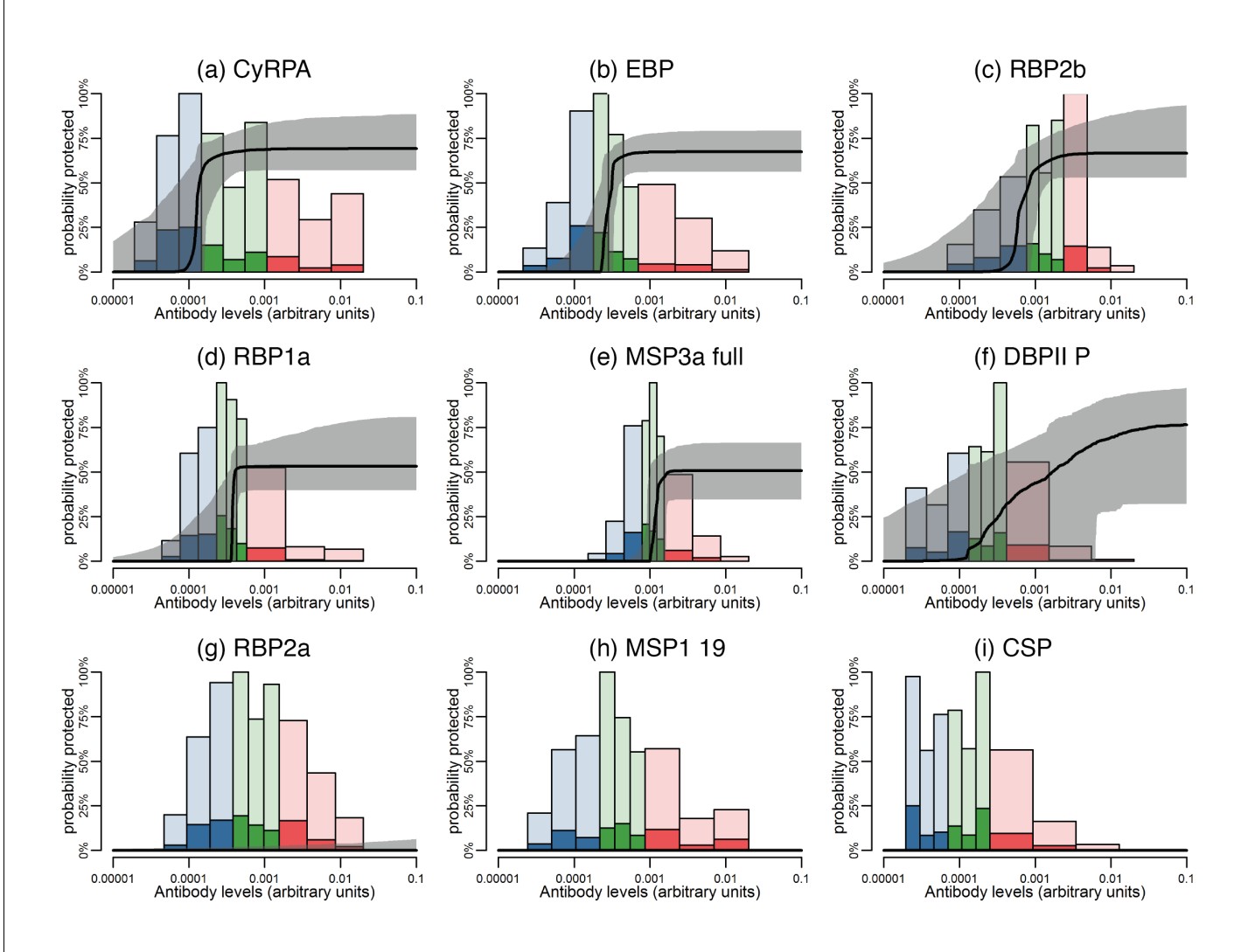

**Figure 6.** Estimated dose-response curves for the associations between antibody responses specific to *P. vivax* antigens and protection from clinical malaria. Solid black lines depict exposure-adjusted dose-response curves, and the grey shaded regions depict the 95% credible intervals. Histograms show the observed distribution of antibody levels (relative to the PNG immune pool) colored per tertiles (low = blue; medium = green; high = red), and the darkly-colored portions of the histograms show the proportion of individuals with that antibody level who had a *P. vivax* episode (>500 parasites/µ L). (**a–c**) Examples of antigens that need low antibody levels to provide 50% of protection. (**d–f**) Examples of antigens that need medium antibody levels to provide 50% of protection. (**g–i**) Examples of antigens that need high antibody levels to provide 50% of protection.

DOI: https://doi.org/10.7554/eLife.28673.011

The following source data and figure supplement are available for figure 6:

**Source data 1.** Antibody levels and 50% protection from clinical malaria.

DOI: https://doi.org/10.7554/eLife.28673.013

**Figure supplement 1.** Estimated dose-response curves for the associations between antibody levels to *P. vivax* antigens and protection from clinical malaria.

DOI: https://doi.org/10.7554/eLife.28673.012

concurrent *P. vivax* infections (n = 30), and 29 increased with age and/or lifetime exposure, indicating the antibody levels are both a marker of concurrent and past exposure to *P. vivax*. Despite these generally low IgG antibody levels, high levels to 31 of 38 proteins (81.6%) were significantly associated with protection, indicating that these antibody levels have started to exceed the threshold above which antibody levels are biomarkers of protection rather than exposure (***Stanisic et al., 2015***).

Antibody levels were often moderately or even highly correlated between antigens, and not only to different regions of the same antigen, such as MSP3α (rho = 0.52–0.93), or to different alleles of the same antigen such as PvDBP (rho = 0.76–0.995). In particular, antibodies to the WGCF-expressed, highly-immunogenic proteins showed very high correlations (rho = 0.61–0.95). Without further functional investigation, it is difficult to determine if these correlations indicate co-acquisition or cross-reactivity, or whether cross-reactive antibodies are functional or merely recognize different proteins.

The high correlations of antibody levels to different DBP alleles were expected, as some are very similar in sequence (e.g. the variants AH and C differ only at amino acid residue 371), thus antibody responses are likely to be highly cross-reactive between variants. However, despite 50% identity, correlations between antibodies to the different alleles of DBPII and EBP were only moderate (rho = 0.36–0.48, p<0.001) and not higher than those with other merozoite surface proteins, indicating only a limited potential for cross-reactive antibodies between these proteins. For the RBP proteins included in this study, we have previously showed that polyclonal antibodies raised in rabbits recognize the homologous but not heterologous proteins (*França et al., 2016b*). The correlated responses are therefore likely due to co-acquisition, and the combination of several RBPs may have synergistic or additive effects (*França et al., 2016b*). For DBPII, although this and previous studies observed the presence of strain-specific antibodies (*Cole-Tobian et al., 2009*), strain-transcending antibodies (in particular those targeting the binding pocket) have been described (*King et al., 2008*; *Chen et al., 2016*; *Wongkidakarn et al., 2016*), indicating that a DBPII vaccine targeting these strain-transcending epitopes may be able to overcome the large genetic diversity of DBPII. Nevertheless, the analyses of antigen combinations clearly indicated that combining DBPII with other antigens such as one of several RBPs is likely to be beneficial.

The majority of the antigens associated with protection against vivax malaria in this cohort were newly described or understudied, such as EBP, P41, CyRPA, RBP1a, and RBP2b. Many of these proteins were similarly or more strongly associated with protective immunity than antibodies to well-studied vaccine candidate antigens such as MSP3α, AMA1, MSP1, and CSP (*Bennett et al., 2016*; *Cutts et al., 2014*; *Stanisic et al., 2013*), clearly demonstrating that the repertoire of possible antigenic targets present in the *P. vivax* genome should be further expanded. Interestingly, among the highly-immunogenic proteins expressed in the WGCF system, only two, PVX_122805, a conserved hypothetical protein and PVX_088820, a member of the Pv-fam-a tryptophan-rich antigen family showed substantial association with protection individually, and were regularly included in combinations of 5–15 antigens. This indicates that immunogenicity by itself is a relatively poor predictor of association with protection, and while such proteins may be good markers of exposure, it is less clear whether they would make effective vaccine candidate antigens.

There was a clear hierarchy in the order in which antigens were included into the best antigen-combinations. The top protective antigens were consistently at the top end of all possible random five-way combinations, an observation that is unlikely to be due to chance if all antigens were equally protective when in combinations. The best single antigen EBP followed by PVX_081550 were always present among the most frequently included proteins in higher combinations. Additional proteins common in combinations of three were CyRPA, RBP1a and RBP2b. Apart from RBP2b, these same proteins are also commonly found in four and five antigen-combinations, together with DBPII, P41 and PVX_099930 (RhopH2).

Literature on the little-studied *P. vivax* antigens associated with protection here is at best scarce. We have recently determined that RBP2b binds exclusively to reticulocytes, raising the prospect that RBP2b may be involved in *P. vivax* reticulocyte-specificity (*França et al., 2016b*). Among the other antigens strongly associated with protection, P41 is a GPI-anchored protein localized on the merozoite surface which forms a heterodimer with P12 (*Hostetler et al., 2015*). In *P. falciparum,* neither P41 nor P12 homologs are essential for in vitro parasite growth (*Tonkin et al., 2013*), but both are immunogenic, and antibodies have been associated with clinical protection (*Richards et al., 2013*; *Osier et al., 2014a*). The *P. falciparum* homolog of PVX_081550 has recently been identified as StAR-related lipid transfer protein localized on the parasitophorous vacuole (*van Ooij et al., 2013*). Both the *P. falciparum* homolog and *P. vivax* PVX_081550 proteins are immunogenic (*Fan et al., 2013*). It is, however, unclear whether antibodies to PVX_081550 interfere with parasite function or are simply elicited by proteins released from the parasitophorous vacuole upon schizont rupture and thus serve as markers of an individual's immune status (*França et al., 2016a*). Lastly, also in *P.*

*falciparum,* CyRPA forms a complex with PfRh5 and PfRipr on the merozoite surface. This complex is involved in binding basigin on the erythrocyte surface (*Reddy et al., 2015*). *P. falciparum* CyRPA seems to not be under immune selection pressure and is highly-conserved, with only a single polymorphism detected across 18 *P. falciparum* strains (*Beeson et al., 2016*). Antibodies raised against *P. falciparum* CyRPA were nevertheless capable of inhibiting merozoite invasion (*Beeson et al., 2016*), block binding of CyRPA to Rh5 and Rh5/CyRPA/Ripr complex formation (*Chen et al., 2017*), and both in vitro and in vivo parasite growth inhibitory activity, more stronger when in combination with PfRh5 (*Favuzza et al., 2017*). The *P. vivax* CyRPA seems to be highly-reactogenic (*França et al., 2016a*), suggesting that their function, or at least exposure to the immune system, may differ between the two species.

IgG subclass responses to some *P. vivax* antigens such as RBP1a, DBPII, MSP1 19, and CSP (*França et al., 2016b*; *Tran et al., 2005*; *Maestre et al., 2010*; *Yildiz Zeyrek et al., 2011*; *Zeyrek et al., 2008*), as well as several different *P. falciparum* antigens (*Stanisic et al., 2015*; *Ahmed Ismail et al., 2014*; *Richards et al., 2010*; *Reiling et al., 2010*; *Noland et al., 2015*; *Tongren et al., 2006*) have consistently shown that the presence of IgG1 and/or IgG3 in variable ratios is likely to play a role in protection against infection and/or clinical disease. The role of IgG2 and IgG4 antibodies however, remains mostly unclear. IgG2 antibodies have been correlated with lower *P. falciparum* parasitemia (*Ahmed Ismail et al., 2014*) and risk of infection (*Aucan et al., 2000*), while IgG4 levels were associated with an enhanced risk of infection and disease (*Aucan et al., 2000*). Neither showed significant ability to promote opsonic phagocytosis (*Osier et al., 2014b*), with IgG4 possibly inhibiting this process (*Chaudhury et al., 2017*). We have previously measured IgG subclass 1, 2, 3 and 4 responses to the 5 RBPs included in this study, showing that IgG1 and IgG3 are the predominant subclasses in this cohort of PNG children (*França et al., 2016b*). IgG1 to RBP1a and RBP2b showed the strongest association with protection in multivariate models. Interestingly, adults showed substantially higher levels of IgG3 for all antigens (IgG3 being predominant for RBP1a), and substantial levels of IgG2 for some (e.g. RBP2b) but not all antigens. Children showed some early evidence of switching to IgG3 for RBP1a and RBP2-P2 with maturation of immune responses, and increase in age and exposure to malaria parasites (*França et al., 2016b*).

Little is known about the threshold necessary for antibody levels to switch from being biomarkers of exposure to biomarkers of protection (*Stanisic et al., 2015*). Large variation in the threshold levels associated with 50% reduction in vivax malaria risk was observed for the present antigen panel. Although this type of analysis cannot differentiate between direct, causal protective effect of antibodies and a statistical association, the observations that many of the antigens most consistently associated with protection in multi-antigen combinations (e.g. PVX_081550, EBP, CyRPA) show both relatively low immunogenicity and low 50% protection threshold levels suggests that for some of the most promising candidate antigens, low antibody levels may be sufficient for protection. It is possible that critical antigens or epitopes have to be weakly immunogenic to avoid acquisition of host immunity. Antigens with little reactivity and associated with acquired immunity at low antibody levels might thus be the most promising for vaccine development. It was, however, not possible to determine in this study whether antibodies to these proteins have better avidity/affinity than some antigens that induce very high antibody levels. One limitation of the dose-response model used in this study is that it only includes one antigen at a time, and thus doesn't account for the complexity of highly-correlated antibody responses.

We focused on 1–3 years old children as early infancy is a critical time in the development of immunity to malaria (*Longley et al., 2016*). Nevertheless, further studies including older age groups from different transmission settings will be required to confirm the detected hierarchy in antibody responses and their association with protection. To our knowledge, the present study is the largest comprehensive panel of *P. vivax* antigens measured in a well-designed cohort study for identification of associations with malaria risk, and the first one to investigate both the effect of multiple combinations and the threshold antibody levels necessary for protection. Our results identify important targets of naturally-acquired immunity that may contribute to clinical protection and should thus be prioritized for functional studies. These results clearly indicate that examining multiple antigens simultaneously can best correlate with protection, as it might optimally define the breadth of the immune response required for clinical immunity. Taken together, our results suggest that a well-designed, multicomponent *P. vivax* vaccine may be more efficacious than single-component vaccines.

Future research should focus on uncovering the functional role of the unknown and understudied proteins identified in this study, and their potential for vaccines, biomarkers of immunity, or both. The measurement of functionality, e.g. antibodies' capacity to promote opsonic phagocytosis or ability to block binding/invasion to red blood cells will be particularly important (*Teo et al., 2016*). Although it remains unclear which mechanisms are most important for protection from malaria infection or clinical disease, recent studies with *P. falciparum* suggest that a combination of functional mechanisms may be important for protective immunity (*Osier et al., 2014b*; *Hill et al., 2013*; *Hill et al., 2016*; *Chiu et al., 2015*; *Joos et al., 2010*; *Murungi et al., 2016*). Hence, the presence/ levels of functional antibodies might possibly be even better correlates of protective immunity than total IgG antibody levels.

## Materials and methods

### Study participants and ethical approval

Samples from a longitudinal cohort of young PNG children were used (*Lin et al., 2010*). Briefly, 264 children aged 1–3 years from an area near Maprik, East Sepik Province, were enrolled from March to September 2006. Children were followed for up to 16 months, with active monitoring of morbidity every two weeks. All PCR-positive *P. vivax* infections were genotyped, allowing the determination of the incidence of genetically distinct blood-stage infections acquired during follow-up (i.e. the molecular force of blood-stage infections, molFOB) (*Koepfli et al., 2013*). Samples collected at the start of the study from 225 children who completed follow-up were included in the present study. Ethical clearance for this study was obtained from the Medical Research and Advisory Committee of the Ministry of Health in PNG (MRAC 05.19), and the Walter and Eliza Hall Institute (HREC 07/07). Written informed consent was obtained from the parents or guardians of all children participating in the PNG cohort study prior to enrolment.

### Antigen selection

A panel of 38 recombinant *P. vivax* antigens was included in this study. The complete list of antigens and their accession numbers can be found in *Supplementary file 1*. Both leading vaccine candidate proteins (i.e. five allelic variants of DBPII, CSP, AMA1 and MSP1), and proteins that are known or predicted to be involved in erythrocyte invasion were included (five different constructs of MSP3α, MSP9, ARP, GAMA, P12, P41, CyRPA, RBP1a, RBP2a, RBP2b, RBP2cNB [i.e. not containing the binding domain], RBP2-P2, and EBP; *Supplementary file 1*). Additionally, bioinformatics approaches were used to identify novel proteins on the basis of expression profile, signal peptide, putative GPI anchor, and homology to known *P. falciparum* antigens (which were thus likely to be exposed to the immune-system during erythrocyte invasion or in schizonts (*Hostetler et al., 2015*; *Arumugam et al., 2014*) (*Supplementary file 1*). These later antigens include both a hypothetical protein thought to be involved in erythrocyte invasion (PVX_081550) and a panel of antigens found to be highly recognized in a screen of plasma from PNG children aged 5–14 years, including RAMA, SERA, RhopH2, AKLP2, CCp5, 2 Pv-fam-a (PVX_088820 and PVX_092995), and hypothetical proteins PVX_080665, PVX_087670, PVX_094350, PVX_114330, and PVX_122085 (Mueller, Takashima and Tsuboi, personal communication).

Antigens were produced in different collaborating laboratories using mostly HEK 293E cells, *E. coli,* or a WGCF expression system as previously described (*Hostetler et al., 2015*; *Gruszczyk et al., 2016*; *Arumugam et al., 2014*). Purification tags included either hexa-his or Cd4 (*Supplementary file 1*). Protein folding has been validated previously by demonstrating that plasma from naturally-exposed populations recognize the native protein, and antibodies to some of them are strongly associated with reduced risk of vivax malaria (*Lu et al., 2014*; *Stanisic et al., 2013*; *Cole-Tobian et al., 2009*; *Hostetler et al., 2015*; *França et al., 2016a*; *Yadava et al., 2007*); by demonstrating that vaccine-induced antibodies raised against the recombinant protein recognize the native protein (*Bennett et al., 2016*); by showing that the proteins have appropriate biological function through the identification of protein-protein interactions (*Hostetler et al., 2015*); or by obtaining a crystal structure (*Gruszczyk et al., 2016*).

## Antibody measurement

Purified proteins were conjugated onto Luminex Microplex microspheres (Luminex Corp.) as described elsewhere (*Kellar et al., 2001*). The concentration of each protein used to conjugate $2.5 \times 10^6$ beads can be found on *Supplementary file 1*. Bead array assays to measure total IgG were performed as described (*França et al., 2016a*). The assay was extensively validated (*França et al., 2016a*; *França et al., 2016b*; *Dent et al., 2015*) prior to testing samples in singlicate. Samples from Australian donors and a standard curve made of pooled plasma from immune PNG adults (at dilutions ranging from 1:50 to 1:51200) were used as controls on each plate.

## Statistical analysis

The dilutions of the immune pool were fitted as plate-specific standard curves using a 5-parameter logistic regression model. For each sample tested, Luminex median fluorescence intensity (MFI) values were interpolated into relative antibody units based on the parameters estimated from the plate' s standard curve. Relative antibody units ranged from $1.95 \times 10^{-5}$ (i.e., equivalent to 1:51200 dilution of the immune pool) to 0.02 (1:50). To account for the background reactivity to the Cd4-tag (*Hostetler et al., 2015*), antibody levels were re-scaled by using linear regression and IgG responses to the Cd4 tag alone (*França et al., 2016a*).

Associations between antibodies and age were assessed using Spearman's rank correlation tests, and differences by infection status using two-tailed unpaired *t*-test after $\log_{10}$ transformation. To analyze the relationship between IgG levels and prospective risk of *P. vivax* episodes (defined as axillary temperature $\geq$37.5°C or history of fever in preceding 48 hr with a concurrent *P. vivax* parasitemia >500 parasites/µL), negative binomial generalized estimating equation (GEE) models with exchangeable correlation structure and semi-robust variance estimator were used (*Stanisic et al., 2013*; *França et al., 2016a*; *França et al., 2016b*). For this, IgG levels were classified into tertiles and analyses done comparing children with medium and high versus low antibody levels. Children were considered at risk from the first day after the blood sample for active follow-up was taken. The molFOB, representing individual differences in exposure, was calculated as the number of new *P. vivax* blood-stage clones acquired per year-at-risk, and square root transformed for better fit (*Koepfli et al., 2013*). All GEE models were adjusted for seasonal trends, village of residency, age, and individual differences in exposure (molFOB).

## Simulated annealing for investigating combinations of antigens

The number of combinations from a panel of 38 antigens is enormous ($2^{38} \sim 2.7 \times 10^{11}$), making investigation of every combination computationally infeasible. In practice, it is possible to investigate all possible combinations of antigens of size up to five, where there are approximately 500,000 combinations (*Osier et al., 2014a*). Investigation of combinations of more than five antigens can be done through efficient exploration of combination space, focusing on the combinations that have the strongest association with protection against clinical malaria.

A simulated annealing algorithm (*Kirkpatrick et al., 1983*) was used to explore the combination space of 38 antigens to identify combinations with optimal potential protective efficacy, defined as the proportional reduction in cases of *P. vivax* clinical malaria comparing children with high versus low antibody levels. The algorithm was implemented with various constraints on the maximum number of antigens in a combination (e.g. no more than 10 antigens allowed in a combination). For each constraint on the maximum number of antigens, the algorithm was repeated 1000 times.

## Dose-response relationship

A dose-response model (*White et al., 2011*; *Chiu et al., 2016*) was used to investigate associations between incidence of clinical malaria and antibody levels. It was assumed that given an individual's antibody level A, the incidence of clinical malaria can be described as:

$$\lambda(A) = \lambda_0 \left( 1 - P_{max} + P_{max} \frac{1}{1 + \left(\frac{A}{\beta}\right)^\alpha} \right)$$

where: $\lambda_0$ = the incidence of clinical malaria in the absence of antibodies; A = an individual's antibody level; $P_{max}$ = the maximum reduction in incidence due to the antibody under investigation;

α = shape parameter of the dose-response curve; and β = scale parameter of the dose-response curve.

The parameters were estimated by fitting to data on the incidence of clinical malaria within each time period. In interval $j$ of duration $T_j$ the probability that an episode of clinical malaria occurs is:

$$P(A_j) = 1 - e^{-\lambda(A_j)T_j}$$

The likelihood that the dose-response function given the data is:

$$L = \prod_j P(A_j)^{I_j} \left(1 - P(A_j)\right) 1 - I_j$$

where $I_j = 1$ for an individual who experienced a clinical episode in interval $j$; $I_j = 0$ if there was no observed episode during the interval.

The model was fitted to the data in a Bayesian framework using Markov Chain Monte Carlo (MCMC) methods. Parameters were assumed to have uniform prior distributions (*White et al., 2011*; *Chiu et al., 2016*).

## Acknowledgements

We thank all patients and their families for participating in this study. We gratefully acknowledge the large Papua New Guinean team for its help in conducting the fieldwork, Ms Sumana Sharma for her help in expressing proteins, and Dr. Rhea Longley for her thoughtful suggestions.

## Additional information

### Funding

| Funder | Grant reference number | Author |
|---|---|---|
| University of Melbourne | Melbourne International Postgraduate Scholarship | Camila Tenorio França Wen-Qiang He |
| National Health and Medical Research Council | 1092789 | Alan F Cowman |
| National Health and Medical Research Council | Program Grant 1092789 | Ivo Mueller Alan F Cowman |
| Japan Society for the Promotion of Science | JP26253026 | Takafumi Tsuboi |
| Japan Society for the Promotion of Science | JP15H05276 | Takafumi Tsuboi |
| Japan Society for the Promotion of Science | JP16K15266 | Takafumi Tsuboi |
| Australian Research Council | Australian Research Council Future Fellowship | Wai-Hong Tham |
| National Institute of Allergy and Infectious Diseases | Intramural Research Program | Rick M Fairhurst |
| National Institutes of Health | AI063135 | Rick M Fairhurst |
| Wellcome | 098051 | Julian C Rayner |
| Medical Research Council | MR/J002283/1 | Julian C Rayner |
| Medical Research Council | MR/L012170/1 | Julian C Rayner |
| National Institutes of Health | U19AI089686 | Ivo Mueller |
| National Health and Medical Research Council | 1021544 | Ivo Mueller |
| Malaria Eradication Scientific Alliance | | Ivo Mueller |

| National Health and Medical Research Council | Independent Research Institute Infrastructure Support Scheme | Ivo Mueller |
| National Health and Medical Research Council | Senior Research Fellowship 1043345 | Ivo Mueller |

The funders had no role in study design, data collection and analysis, decision to publish, or preparation of the manuscript. The views expressed in this article are those of the authors and do not necessarily reflect the official policy or position of the Department of Health and Human Services, Department of the Army, the Department of Defense, nor the U.S. Government.

### Author contributions

Camila Tenorio França, Formal analysis, Validation, Investigation, Visualization, Methodology, Writing—original draft; Michael T White, Formal analysis, Visualization, Methodology, Writing—original draft; Wen-Qiang He, Investigation, Writing—review and editing; Jessica B Hostetler, Jakub Gruszczyk, Anjali Yadava, Mary R Galinski, Julie Healer, Chetan Chitnis, Eizo Takashima, Resources, Writing—review and editing; Jessica Brewster, Validation, Investigation; Gabriel Frato, Indu Malhotra, Christele Huon, Enmoore Lin, Benson Kiniboro, Resources, Methodology; Peter Siba, Conceptualization, Resources, Funding acquisition; Alan F Cowman, Takafumi Tsuboi, Wai-Hong Tham, Rick M Fairhurst, Julian C Rayner, Resources, Funding acquisition, Writing—review and editing; Christopher L King, Conceptualization, Resources, Funding acquisition, Methodology, Writing—review and editing; Ivo Mueller, Conceptualization, Resources, Data curation, Formal analysis, Supervision, Funding acquisition, Methodology, Project administration, Writing—review and editing

### Author ORCIDs

Camila Tenorio França (iD) http://orcid.org/0000-0001-5885-4410
Jakub Gruszczyk (iD) http://orcid.org/0000-0003-4536-2964
Anjali Yadava (iD) https://orcid.org/0000-0002-6751-954X
Alan F Cowman (iD) http://orcid.org/0000-0001-5145-9004
Ivo Mueller (iD) http://orcid.org/0000-0001-6554-6889

### Ethics

Human subjects: Ethical clearance for this study was obtained from the Medical Research and Advisory Committee of the Ministry of Health in PNG (MRAC 05.19), and the Walter and Eliza Hall Institute (HREC 07/07). Written informed consent was obtained from the parents or guardians of all children participating in the PNG cohort study prior to enrollment.

### Decision letter and Author response

Decision letter https://doi.org/10.7554/eLife.28673.016
Author response https://doi.org/10.7554/eLife.28673.017

## Additional files

### Supplementary files

• Supplementary file 1. *P. vivax* antigens included in this study. Conc = concentration; HEK293E = human embryonic kidney 293E cells; WGCF = wheat germ cell-free. *Cd4-tagged proteins. Cd4 alone was conjugated to Luminex beads (2 μg/ml per $2.5 \times 10^6$ beads) and tested in all samples as a control for background reactivity.
DOI: https://doi.org/10.7554/eLife.28673.014

• Transparent reporting form
DOI: https://doi.org/10.7554/eLife.28673.015

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
