## [Decision Letter]

Thank you for submitting your article "Identification of highly-protective combinations of *Plasmodium vivax* recombinant proteins for vaccine development" for consideration by *eLife*. Your article has been reviewed by three peer reviewers, one of whom is a member of our Board of Reviewing Editors and the evaluation has been overseen by Prabhat Jha as the Senior Editor. The following individuals involved in review of your submission have agreed to reveal their identity: Arturo Reyes-Sandoval (Reviewer #2); Josue Lima-Junior (Reviewer #3).

The reviewers have discussed the reviews with one another and the Reviewing Editor has drafted this decision to help you prepare a revised submission.

Summary:

*Plasmodium vivax* is the most widespread parasite causing malaria. Despite that *P. vivax* has been gaining much needed attention concerning approaches towards its eradication, be it via drugs, vaccines and other means (bed nets, etc.), research surrounding *P. vivax* lags behind *P. falciparum*. Only a limited number of antigens are available for testing as vaccine candidates and in the absence of long-term in vitro *P. vivax* culture techniques, studying immune responses in naturally infected *P. vivax* infected populations constitutes a major way forward towards antigen discovery and eventually development *P. vivax* vaccine.

This very ambitious investigation represents a longitudinal 16-month follow-up study of a cohort of 264 children in PNG. The children were carefully genotyped for *P. vivax* infections and their antibody levels were measured against 38 selected *P. vivax* antigens that represent well-defined proteins as well as new proteins identified by in silico approaches. Using a novel simulated annealing algorithm to explore the potential of antigen combinations in relation to protective efficacy, the authors describe for the first time synergistic effect of combinations of antibody responses directed to multiples *P. vivax* antigens. Having identified combinations of no more than 5 antigens with >90% potential protective efficacy, the authors claim that the synergistic or additive effect of combinations of antibody response to a large panel of *P. vivax* antigens represents a new approach towards development of multicomponent *P. vivax* vaccine.

Essential revisions:

What follows are specific comments provided by the reviewers and the authors are requested to make appropriate revisions to the manuscript for a consideration to be published in *eLife*.

1) Please provide either concentrations or ODs of the pooled sera from adult PNG subjects for a comparison vis-à-vis the "protective" levels in children.

2) IgG responses against *P. falciparum* antigens have consistently shown that while IgG1 and IgG3 play a role in protection, IgG2 and IgG4 might interfere in the acquisition of clinical immunity. In *P. vivax*, these associations present conflicting results. If the IgG subclass profile data specific for the *P. vivax* recombinants are available, please include them.

3) Please comment in the discussion on the possible relationship between IgG subclass prevalence for the known vaccine candidates and the association with protection found in this work.

4) Please explain the reason for the different concentrations of purified recombinant proteins used to couple beads in the Luminex assay. It appears that the low antibody concentration required to achieve protection in the dose response model (subsection “Thresholds of IgG levels and protection”) were directed to 5 out 6 proteins that were used at higher concentrations in the Luminex assay. Do these observations indicate any relationship between the different concentration and the levels of specific antibody?

5) It appears that 3 antigens are equally or nearly as effective for protective efficacy conferred by 5 antigens. The authors are encouraged to offer some comments regarding hierarchical arrangement of their priority antigens on the basis of results indicated in the heatmap.

---

## [Author Response]

Essential revisions:What follows are specific comments provided by the reviewers and the authors are requested to make appropriate revisions to the manuscript for a consideration to be published in eLife.1) Please provide either concentrations or ODs of the pooled sera from adult PNG subjects for a comparison vis-a-vis i the "protective" levels in children.

We used the parameters estimated from plate-specific standard curves to interpolated the median fluorescence intensity (MFI) values from the subject samples into relative antibody unit values that range from 1.95x10^-5^ (i.e., equivalent to 1:51200 dilution of the immune pool) to 0.02 (1:50).

Whereas the use of standard curves made of pooled serum did not give us absolute antibody concentrations, it gave us the ability to directly compare across different subjects and cohort studies. The "protective" levels shown in our dose response models, therefore, refer to a given dilution of the immune pool that serves as a threshold, above which any individual/population can be considered at lower risk of having clinical *P. vivax* episodes.

We believe that the only way to measure the exact concentration of the antigen-specific antibodies present in the pooled serum used is by affinity-purifying them using the each of the antigens. Given the large number of antigens included in this study and the amount of serum that would be necessary to affinity-purify these antibodies, this is, unfortunately, not feasible.

To clarify that our values are in relative antibody units, we have included the following:

Subsection “Statistical analysis”: “The dilutions of the immune pool were fitted as plate-specific standard curves using a 5-parameter logistic regression model. For each sample tested, Luminex median fluorescence intensity (MFI) values were interpolated into relative antibody units based on the parameters estimated from the plate’ s standard curve. Relative antibody units ranged from 1.95x10-5 (i.e., equivalent to 1:51200 dilution of the immune pool) to 0.02 (1:50). To account for the background reactivity to the Cd4-tag (35), antibody levels were re-scaled by using linear regression and IgG responses to the Cd4 tag alone (38).

Subsection “Thresholds of IgG levels and protection”: “Using a mathematical dose-response model adjusted for exposure (molFOB), predicted threshold ranges (relative to the PNG immune pool) at which antibodies become associated with protection were observed for the different antigens (Figure 6; Figure 6—figure supplement 1).

The world “concentration” has been replaced by “level” or “relative antibody level” throughout the manuscript.

We have also changed the legend of Figure 6 from “antibody titres” to “antibody levels (arbitrary units)”.

2) IgG responses against P. falciparum antigens have consistently shown that while IgG1 and IgG3 play a role in protection, IgG2 and IgG4 might interfere in the acquisition of clinical immunity. In P. vivax, these associations present conflicting results. If the IgG subclass profile data specific for the P. vivax recombinants are available, please include them.

We fully agree with the reviewers that it is very important to characterize antibody subclass responses and establish their role in protection against clinical malaria. We have started the analysis of levels of antigen-specific IgG subclasses 1, 2, 3 and 4 for the following antigens:

The members of the PvRBP family (França et al., 2016). We found that IgG1 is the most prevalent IgG subclass in this cohort of young PNG children for all antigens, with significant levels of IgG3 for some (e.g. PvRBP1a) but not all antigens. IgG1 showed the strongest association with protection in multivariate models. In the adult pool, we detected substantially higher levels of IgG3 for all antigens (IgG3 being predominant for PvRBP1a), and substantial levels of IgG2 for some (e.g. PvRBP2b) but not all antigens. This indicates that as immune responses mature and protection increases, there is a switch from IgG1 predominance to increased levels of IgG3, and some IgG2.

We have since completed similar analysis for the PvDBP alleles and PvEBP. These results showed a similar predominance of IgG1 in the young PNG children. These results are being presented for publication in a separate manuscript.

We are now extending these analysis to other proteins with the highest multivariate association with protection (e.g. CyRPA and PVX_081550).

We have included a reference to the published results on the PvRBP family into the discussion, and highlighted the predominance of IgG1 in these young children and the switch to more IgG3 and some IgG2 in adults. Discussion section (see below).

3) Please comment in the discussion on the possible relationship between IgG subclass prevalence for the known vaccine candidates and the association with protection found in this work.

We have now extended the discussion to include references to IgG subclasses and their association with protection for those antigens where such results have been published.

Discussion section: “IgG subclass responses to some *P. vivax* antigens such as PvRBP1a, DBPII, MSP1 19, and CSP (Chen et al., 2016, Yildiz et al., 2011, Zeyrek et al., 2008, Ahmed et al., 2014 and Richards et al., 2010), as well as several different *P. falciparum* antigens (Franca et al., 2016, Reiling et al., 2010, Noland et al., 2015, Tongren et al., 2006, Aucan et al., 2000 and Osier et all., 2014) have consistently shown that the presence of IgG1 and/or IgG3 in variable ratios is likely to play a role in protection against infection and/or clinical disease. The role of IgG2 and IgG4 antibodies however, remains mostly unclear. IgG2 antibodies have been correlated with lower *P. falciparum* parasitemia (Reiling et al., 2010) and risk of infection (Chaudhury et al., 2017), while IgG4 levels were associated with an enhanced risk of infection and disease (Chaudhury et al., 2017). Neither showed significant ability to promote opsonic phagocytosis (Teo et al., 2016), with IgG4 possibly inhibiting this process (Hill et al., 2013). We have previously measured IgG subclass 1, 2, 3 and 4 responses to the 5 PvRBPs included in this study, showing that IgG1 and IgG3 are the predominant subclasses to PvRBPs in this cohort of PNG children. IgG1 to PvRBP1a and PvRBP2b showed the strongest association with protection in multivariate models. Interestingly, adults showed substantially higher levels of IgG3 for all antigens (IgG3 being predominant for PvRBP1a), and substantial levels of IgG2 for some (e.g. PvRBP2b) but not all antigens. Children showed some early evidence of switching to IgG3 for PvRBP1a and PvRBP2-P2 with maturation of immune responses, and increase in age and exposure to malaria parasites (Chen et al., 2016).”

4) Please explain the reason for the different concentrations of purified recombinant proteins used to couple beads in the Luminex assay. It appears that the low antibody concentration required to achieve protection in the dose response model (subsection “Thresholds of IgG levels and protection”) were directed to 5 out 6 proteins that were used at higher concentrations in the Luminex assay. Do these observations indicate any relationship between the different concentration and the levels of specific antibody?

The concentration of each antigen conjugated to the beads was determined so that the entire adult pool standard curve (i.e. from 1:50 to 1:51200) was log-log-linear. Higher antibody concentration on the bead thus reflect a lower concentration of antibodies in the adult pool (taking into account that different proteins are of different lengths and are likely to also contain a variable numbers of B-cell epitopes). As explained above, the levels in children are the relative levels compared to ‘equilibrium’ antibody levels that the immune PNG adults have achieved.

In order to assure the observed results were not influenced by the conjugated antigen concentration, we have now correlated the antigen concentration with the geometric mean antibody levels observed in children, the levels at which 50% protection is achieved (if it is achieved at all), and the maximum level of protection achieved. For none of these variables a significant correlation was observed (Spearman rank correlation range: -0.16 to 0.20, P > 0.3). We are, therefore, confident that the conjugated antigen concentration did not influence the relationship between antibody concentration and the level protection achieved.

These results have been included as Figure 6—figure supplement 2, as well as in the Results section.

Subsection “Thresholds of IgG levels and protection”: In order to assure the observed results were not influenced by the concentration of the antigen conjugated onto the assay beads, we correlated the antigen concentration with the geometric mean antibody levels observed in children, the levels at which 50% protection is achieved (if it is achieved at all), and the maximum level of protection achieved. For none of these variables a significant correlation was observed (rho= -0.16 to 0.20, P>0.3), indicating that the conjugated antigen concentrations did not influence the relationship between antibody levels and the level protection achieved (Figure 6—figure supplement 2).”

5) It appears that 3 antigens are equally or nearly as effective for protective efficacy conferred by 5 antigens. The authors are encouraged to offer some comments regarding hierarchical arrangement of their priority antigens on the basis of results indicated in the heatmap.

The reviewer is correct that there is a diminishing return as higher combination of antigens are tested.

**1 antigen****2 antigens****3 antigens****4 antigens****5 antigens**# combinations38703843673815501942median PPE37.1%48.1%55.1%59.8%63.4%maximum PPE63.4%79.3%82.8%84.4%85.9%

While the medium PPE is increasing strongly from 37.1% for a single antigen to 55.1% for 5, and to 63.4% for 5 antigen combinations, this trend is even more evident when the maximum PPE is considered (see table above).

However, there was a clear hierarchy in the order in which antigens were being included into the best antigen combination. The best single antigen EBP was always present among the most frequently included proteins of higher combinations. The 2^nd^ protein was PVX_081550. Additional proteins common in combinations of three were CyPRA, RBP1a and RBP2b. Apart from RBP2b, these same proteins were also commonly found in four and five antigen-combinations, together with DBP, P41 and PVX_099930 (RhopH2).

We have now included in the Discussion section a paragraph on the hierarchy of antigen inclusions in more and more complex combinations.

Discussion section: “There was a clear hierarchy in the order in which antigens were included into the best antigen-combinations. The top protective antigens were consistently at the top end of all possible random five-way combinations, an observation that is unlikely to be due to chance if all antigens were equally protective when in combinations. The best single antigen EBP followed by PVX_081550 were always present among the most frequently included proteins in higher combinations. Additional proteins common in combinations of three were CyRPA, RBP1a and RBP2b. Apart from RBP2b, these same proteins are also commonly found in four and five antigen-combinations, together with DBPII O, P41 and PVX_099930 (RhopH2).”